# Digitalization of Clubroot Disease Index, a Long Overdue Task

**Rasha Salih [1] and Edel Pérez-López [1,2,\*]**

1.  Department of Plant Sciences, Faculté des Sciences de l'Agriculture et de l'Alimentation (FSAA), Université Laval, Quebec City, QC G1V 0A6, Canada; rasha.salih.1@ulaval.ca
2.  Centre de Recherche et D'Innovation sur les Végétaux (CRIV), Université Laval, Quebec City, QC G1V 0A6, Canada
\*   Correspondence: edel.perez-lopez@fsaa.ulaval.ca

**Abstract:** Clubroot is a devastating disease caused by the protist *Plasmodiophora brassicae* Woronin. After root hair colonization, the clubroot pathogen induces clubs that block water uptake, leading to dehydration and death. The study of the severity of plant diseases is very important. It allows us to characterize the level of resistance of plant germplasm and to classify the virulence of pathogen strains or isolates. Lately, the use of learning machines and automatization has expanded to plant pathology. Fast, reliable and unbiased methods are always necessary, and with clubroot disease indexing this is not different. From this perspective, we discuss why this is the case and how we could achieve this long overdue task for clubroot disease.

**Keywords:** *Plasmodiophora brassicae*; cruciferous crops; image-based indexing; machine learning; clubroot disease

## 1. Clubroot, an Introduction

In recent years, clubroot has become one of the most devastating diseases affecting the family *Brassicaceae* worldwide. This disease is caused by the soil-borne obligate parasite *Plasmodiophora brassicae* [1]. The susceptible plant hosts develop clubs on their roots as a result of *P. brassicae* infection, leading to dehydration and death of the infected plant [1]. Clubroot disease usually leads to approximately 10–15% yield losses on a global scale and could exceed 30% under disease-inducing environmental conditions [2,3]. Control of clubroot disease has proven to be very difficult to achieve, mainly because the resting spores released into the soil by the pathogen are resistant to most of the fungicides available on the market, and the fact that they can survive on the soil for many years [1]. To this day, the best alternative is the use of resistant genotypes, but after one or two years of using certain clubroot-resistant crops, resistance has been broken by different *P. brassicae* pathotypes [4]. A very important step to identify which crop variety is the best for each field is the determination of the virulence of the different isolates in susceptible and clubroot-resistant germplasm. This relies on the determination of disease index (DI) through a very laborious observational and subjective process [4–9]. Disease indexing would benefit from digitalization, which not only would make it easier and faster, but also reproducible across different laboratories. From this perspective, we discuss why this is the right time for the digitalization of clubroot disease indexing and how we could be able to achieve it successfully.

## 2. Clubroot Disease Index and *Plasmodiophora brassicae* Pathotyping

The management of clubroot disease relies on the use of clubroot-resistant (CR) varieties. However, the genetic resistance to clubroot is highly vulnerable to pathotype shifts. Currently, the use of host differential systems along with population genetics provides insights into the pathogen identity [4]. The most commonly employed differentials to identify *P. brassicae* pathotypes are Somé [5], Williams [6], European Clubroot Differential (ECD) [7] and the Canadian Clubroot Differential (CCD) [8]. To date, more than

30 *P. brassicae* pathotypes have been identified in Canada using the CCD [8]. Although this differential system has been effective for detecting predominant pathotypes of *P. brassicae* in Canadian fields and to study the virulence of single spore isolates (SSI), it does not discriminate between virulent and avirulent strains of *P. brassicae* against the resistance in CR varieties or cultivars. This has been addressed by the novel differential system SCD (Sinitic Clubroot Differential), which theoretically has the potential to detect around 250 pathotypes and has been successful in different isolates from *Brassica* crops in China and Korea [9].

Disease indexing is a key part of the classification of *P. brassicae* pathotypes, allowing us to evaluate the virulence of the pathogen. Usually, after 35–45 days post inoculation of the pathogen the disease symptoms are assessed. The commonly accepted practice is ranking the root for the severity of the disease by the human eye. The scoring system for the galls in the roots follows a score of 0 to 3, where 0—no symptoms, 1—few small clubs on lateral roots, 2—small clubs on the main root and larger clubs on lateral roots and 3—large galls on both main root and lateral roots.

Although ranking the infected roots via visual estimation by trained personnel is the current way of calculating the disease index of clubroot, it is inefficient in terms of labor and cost requirements and could be subject to human biases as well. In addition, the assessments can be influenced by temporal variations as well as those acquired from different assessors. Therefore, the foregoing drawbacks demand an objective visualization technique with the highest accuracy and reproducibility.

### 3. Moving to the Digital World—Which Are the Best Alternatives?

Lately, digital imaging systems are being adopted across many scientific disciplines for rapid pathotyping and disease indexing in plant science [10]. The digitalization of root morphology and architecture is often achieved by obtaining digital root images followed by analyses using software such as WinRhizo, ROOTEDGE, RootSystemAnalyser, GLO-RIA, RootGraph, RhizoScan and others [11]. These software are broadly grouped as semi-automated construction, fully automated reconstruction and fully automated phene construction, depending on the architecture of the roots. A comprehensive description of the foregoing software packages can be found in the following online resource (www.plant-image-analysis.org) (accessed on 1 July 2021). These software aid in assessing the number of roots, number of root tips, depth, width, depth–to-width ratio, areas, angles and other topological features [12–14].

Selecting the appropriate growing conditions of the sample population is essential in selecting the suitable digital software, because obtaining the right images for analysis is a trade-off between the growth environment and its throughput. Therefore, choosing the right growth conditions is challenging as there are many methods available (Table 1). When selecting the method, care should be taken to recreate as closely as possible the same environment in which the plants are infected in the field. The *P. brassicae* infections are conditioned by the transition of resting spores into motile spores, zoospores, which are chemotactically attracted by the host roots under water-saturated conditions [1]. In fact, hydroponic systems have been already successfully applied to grow *Brassica* spp. and to infect the plants with the clubroot pathogen resting spores [15], but several modifications should be performed in order to evaluate the infected roots with any of the software mentioned above.

**Table 1.** Plant growth conditions and pros/cons for further root analyses.

| Method | Pros | Cons | Reference |
| --- | --- | --- | --- |
| Root excavations and trenching | Natural conditions<br>Restrictions to growth<br>Complete lifespan of plants<br>3D growth environment<br>Soil conditions | Destructive<br>Time consuming | [16] |
| Shovelomics | Natural conditions<br>Restrictions to growth<br>Complete lifespan of plants<br>3D growth environment<br>Soil conditions | Destructive<br>Only a part of the root system is analysed | [17] |

<div align="center">

**Table 1.** *Cont.*

</div>

| Method | Pros | Cons | Reference |
|---|---|---|---|
| Field minirhizotrons | Soil conditions<br>Natural atmospheric conditions<br>Complete lifespan of plants | Only part of the root system is observable | [18] |
| Hydroponic system | Easy and direct access for the roots<br>Uniform conditions<br>Complete lifespan of plants<br>3D growth environment | No physical constraints to growth | [19] |
| Aeroponic growth | Easy to move the samples in the system<br>Easy and direct access for the roots<br>Uniform conditions | No physical constraints to growth<br>Artificial soil environment | [16] |
| Growth on filter paper pouches | Easy to handle<br>Clear difference between the filter paper and the roots | Contamination by fungi<br>Artificial root environment<br>No physical constraints to growth<br>2D growth<br>Shorter cultivation time | [20] |

## 4. Conclusions

Climate change is impacting clubroot distribution and new geographic areas are becoming affected by the disease, along with new and uncharacterized *P. brassicae* pathotypes with increased virulence towards resistant canola cultivars [21]. There is a long way to go before fully automated clubroot disease indexing (Figure 1), but we believe that it is and will be, in the near future, extremely necessary.

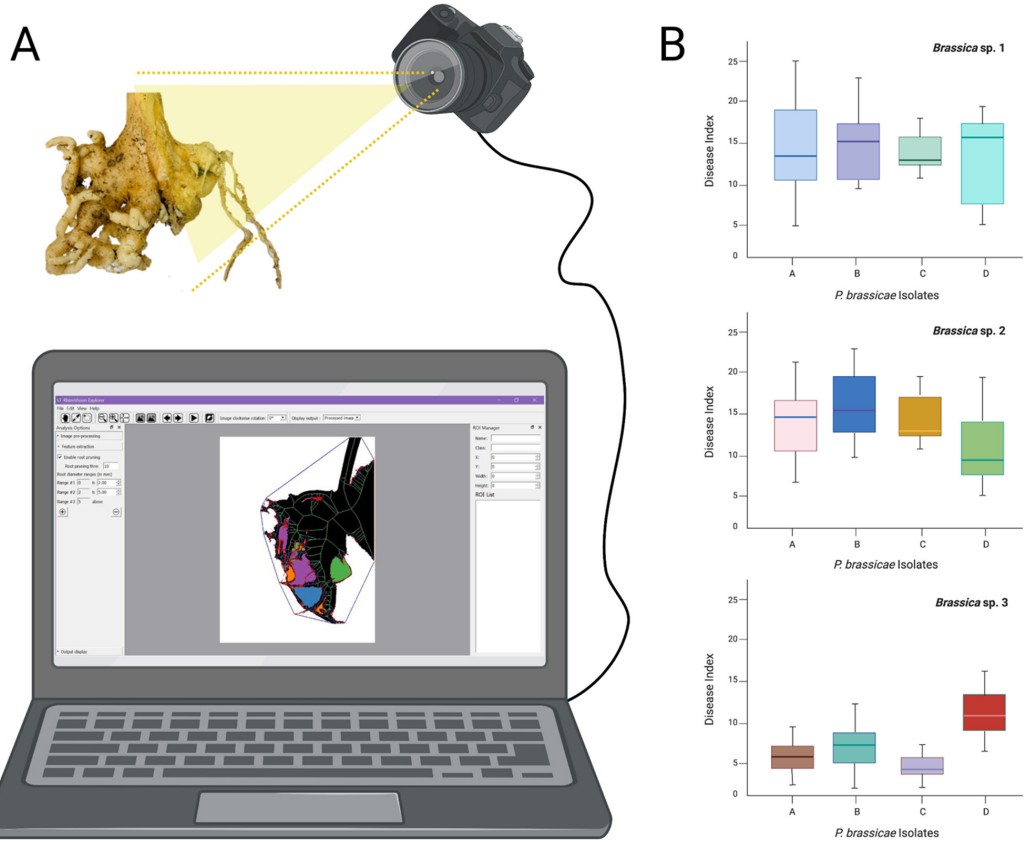

**Figure 1.** Schematic representation of root galls digitalization. (**A**), analysis of roots using a suitable software and disease index for *P. brassicae* isolates infecting different canola genotypes. (**B**), the graphics presented are hypothetical, made with three hypothetical *Brassica* spp. (1 to 3) and four hypothetical clubroot pathogen isolates (A to D), used to represent how we could present the clubroot disease index calculated by the software.

**Author Contributions:** Conceptualization, R.S. and E.P.-L.; writing—original draft preparation, R.S. and E.P.-L.; writing—review and editing, R.S. and E.P.-L.; supervision, E.P.-L.; funding acquisition, E.P.-L. All authors have read and agreed to the published version of the manuscript.

**Funding:** This research was funded by Canola Council of Canada and Western Grains Research Foundation, grant number 2021.04.

**Institutional Review Board Statement:** Not applicable.

**Informed Consent Statement:** Not applicable.

**Conflicts of Interest:** The authors declare no conflict of interest.

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
