# Peer review of "Digitalization of Clubroot Disease Index, a Long Overdue Task"

_horticulturae, doi:10.3390/horticulturae7080241_

Round 1
Reviewer 1 Report
Congratulations for the scientific paper written by you!
Author Response
Dear reviewer,
Thank you very much for the positive comments.
Reviewer 2 Report
all information in the attached file

Author Response
Dear reviewer,
Thank you very much for the comments.
- The authors, unfortunately, did not give a strong evidence base, based on their own or other researchers' data.
Response: We would like to clarify that this article is a perspective, where we are trying to introduce to the community how important would be the digitalization of the clubroot disease indexing. We are sorry for mentioning that we tested one of the software. Perspectives should not include original data. To ensure clarity we have removed that section from the document.
2. Is Fig.1 the original data of the authors or taken from literary sources? If it is the original data, is it necessary to include a discussion, because it is not clear which of the software was used to determine the disease index and how it corresponds to the classical visual assessment? What was meant by genotypes (sample or individual plants)? What is meant by isolates A, B, C, D? If data from other authors is used, references should be made to the relevant studies.
Response: The Figure 1 is a schematic representation of how it would look the analysis of the clubs with one of the software proposed. The graphs are mock also representing a real analysis. We have changed canola genotype by Brassica sp. 1 to 3. We have clarified that now in the new legend: "Figure 1. Schematic representation of root galls digitalization, analysis of roots using a suitable software and disease index for P. brassicae isolates infecting different canola genotypes. The graphics presented are hypothetical made with three hypothetical Brassica spp. (1 to 3) and four hypothetical clubroot pathogen isolates (A to C) used to represent how we could present the clubroot disease index calculated by the software."
3. The article deals with the pathogen Plasmodiophora brassicae (plasmodium), and Table 1 compares the methods of infecting plants from different families and pathogens of other levels of organization: Phytophthora sojae (oomycete) and Phymatotrichopsis omnivora (micromycete). They all have different relations in the "plant-parasite" system, and, there will be absolutely different methods of infection, symptoms of manifestation and accounting, depending on the technology of cultivation of host plants. Therefore, the conclusion of authors "...Therefore, a hydroponic system could provide the ideal condition to initiate and observe infections..." is incorrect (section "Moving to the digital world - which are the best alternatives?"). We propose the authors to consider in Table 1 different techniques for Plasmodiophora brassicae infestation.
Response: We have removed the previous conclusion. The idea with Table 1 is to highlight that the selection of a good method to grow the plants is very important in order to be able to digitalize the infected and non-infected roots. The infection with P. brassicae is very simple, basically consist in applying the resting spores to the substrat. We have added the following from line 85-89: "In fact, hydroponic systems have been already successfully applied to growth Brassica spp. and to infect the plants with the clubroot pathogen resting spores [16], but several modifications should be performed in order to evaluate the infected roots with any of the software mentioned above." We would like to keep Table 1 as it is because it would be of great help for the community trying to improve clubroot disease indexing.
We hope the corrections done through the manuscript make it more clear and easy to understand. We have attached the version with the tracks to show the rest of the modifications and clarifications performed.
Thank you for agreeing to review the manuscript and looking forward to seeing this work published in Horticulturae.
Edel Pérez-López

Reviewer 3 Report
The perspective looks interesting. It can be published if the authors answer the following comment -
Out of all available software, why did the authors only evaluate the use of RhizoVision Explorer?
Author Response
Dear reviewer,
Thank ou very much for the comments and suggestions. We have performed substantial editions of the manuscript (see version attached).
1. Out of all available software, why did the authors only evaluate the use of RhizoVision Explorer?
Response: Analyzing the document now we noticed that the inclusion of that section would go against the rules of a Perspective document because unpublished results should be avoided. Now we have removed that from the document. The answer of why we used it is simply that we wanted to have a prove of concept but in the future other software should be investigated.
Thank you again for the comments,
Edel Pérez-López

Round 2
Reviewer 2 Report
The authors have taken all our comments into account and improved the manuscript. The article is of an advertising rather than scientific nature. If this fact does not violate the rules of publication in the journal, it can be recommended for publication in this form.